# Asymptotic Expansions for Symmetric Statistics with Degenerate Kernels

**Shuya Kanagawa** [1,2]

1 Department of Mathematics, Tokyo Gakugei University, 4-1-1 Nukuikita-machi, Tokyo 184-8501, Japan; skanagaw@tcu.ac.jp
2 Department of Mathematics, Tokyo City University, 1-28-1 Tamazutsumi, Tokyo 158-8557, Japan

**Abstract:** Asymptotic expansions for U-statistics and V-statistics with degenerate kernels are investigated, respectively, and the remainder term $O(n^{1-p/2})$, for some $p \geq 4$, is shown in both cases. From the results, it is obtained that asymptotic expansions for the Cramér–von Mises statistics of the uniform distribution $U(0,1)$ hold with the remainder term $O\left(n^{1-p/2}\right)$ for any $p \geq 4$. The scheme of the proof is based on three steps. The first one is the almost sure convergence in a Fourier series expansion of the kernel function $u(x,y)$. The key condition for the convergence is the nuclearity of a linear operator $T_u$ defined by the kernel function. The second one is a representation of U-statistics or V-statistics by single sums of Hilbert space valued random variables. The third one is to apply asymptotic expansions for single sums of Hilbert space valued random variables.

**Keywords:** U-statistics; V-statistics; asymptotic expansion; integral kernel; nuclearity

**MSC:** 60B12; 60F05; 62G20

## 1. Introduction

Asymptotic expansions for symmetric statistics are studied by many people. See, e.g., Callaert–Janssen–Veraverbeke (1980) [1], Withers (1988) [2], Maesono (2004) [3], and so on. They treat U-statistics with non-degenerate kernels. On the other hand, Bentkus—Götze (1999) [4]and Zubayraev (2011) [5] obtained optimal bounds in asymptotic expansions for U-statistics with degenerate kernels. They treat the following modified U-statistics,

$$W_n = \frac{1}{n^2} \sum_{1 \leq i < j \leq n} \phi(\xi_i, \xi_j) + \frac{1}{n} \sum_{1 \leq i \leq n} \phi_1(\xi_i), \tag{1}$$

where $\phi(\cdot, \cdot)$ is a symmetric function, $\phi_1(\cdot)$ is a measurable function and $\{\xi_i\}$ are i.i.d. random variables. $W_n$ coincides with V-statistics when

$$\phi_1(x) = \frac{1}{2}\phi(x, x). \tag{2}$$

If $\phi_1(x) = 0$ for any $x$, then $W_n$ coincides with U-statistics. They obtained asymptotic expansions with remainder $O(n^{-1})$ for the distribution function of $W_n$. In this paper, we investigate asymptotic expansions for the simple U-statistics and the V-statistics with degree two defined by

$$U_n = \frac{2}{n^2} \sum_{1 \leq i < j \leq n} u(\xi_i, \xi_j), \quad V_n = \frac{1}{n^2} \sum_{1 \leq i, j \leq n} u(\xi_i, \xi_j), \tag{3}$$

respectively. We obtain asymptotic expansions with remainder $O(n^{1-p/2})$ for some $p \geq 4$ for the distribution function of $U_n$ or $V_n$ under some assumptions for $\{\xi_i\}$ and $u(x,y)$. Our scheme of the proof is based on three steps. The first one is the almost sure convergence

in a Fourier series expansion of $u(\xi_i, \xi_j)$. The key condition for the convergence is the nuclearity of a linear operator $T_u$ defined by the kernel function $u(x, y)$. The second one is a representation of U-statistics or V-statistics by single sums of Hilbert space valued random variables. The third one is to apply asymptotic expansions for single sums of Hilbert space valued random variables due to Sazonov—Uyanov (1995) [6].

## 2. Symmetric Statistics

Let $\{\xi_j,\ j \geq 1\}$ be i.i.d. random variables with a probability distribution $\mu$ on an arbitrary measurable space $(X,\ \mathcal{B})$. Suppose that $u(x_1, x_2, \cdots, x_n)$ is a real valued symmetric function for some $k \geq 1$, i.e.,

$$u(x_1, x_2, \cdots, x_k) = u(x_{i_1}, x_{i_2}, \cdots, x_{i_k}), \tag{4}$$

for any permutation $(i_1, i_2, \cdots, i_k)$ of $(1, 2, \cdots, k)$. A statistics defined by the kernel function $u(x_1, x_2, \cdots, x_k)$ is called a symmetric statistics. The followings are the typical examples of the symmetric statistics.

**Example 1.** *U-statistics with degree $k \geq 1$:*

$$U_n = \binom{n}{k}^{-1} \sum_{1 \leq i_1 < i_2 < \cdots < i_k \leq n} u(\xi_{i_1}, \xi_{i_2}, \cdots, \xi_{i_k}). \tag{5}$$

**Example 2.** *V-statistics with degree $k \geq 1$:*

$$V_n = n^{-k} \sum_{1 \leq i_1, i_2, \cdots, i_k \leq n} u(\xi_{i_1}, \xi_{i_2}, \cdots, \xi_{i_k}). \tag{6}$$

In this paper, we treat V-statistics $V_n$ and U-statistics $U_n$ with degree two defined by (3) when the kernel function $u(x, y)$ is degenerate, i.e.,

$$E[u(\xi_1, x)] = 0, \tag{7}$$

for any real number $x$.

## 3. Non-Central Limit Theorems for U-Statistics with Degenerate Kernels

Assume that $\{\xi_i\}$ are i.i.d. random variables with a distribution $\mu$. Let $u(x, y)$ be a real valued symmetric function on $\mathbf{R} \times \mathbf{R}$ and square integrable such that

$$E\left[u(\xi_1, \xi_2)^2\right] < \infty. \tag{8}$$

Suppose that $u(x, y)$ is a degenerate kernel satisfying the condition (7). Let $L^2(\mathbf{R}, \mu)$ be the space of all square integrable functions with respect to $\mu$. Then, according to Serfling (1980) [7], we see that the kernel $u(x, y)$ induces a bounded linear operator $L^2(\mathbf{R}, \mu) \to L^2(\mathbf{R}, \mu)$ (trace class) defined by

$$T_u(f) = E[u(\xi_1, x)f(\xi_1)] = \int_{-\infty}^{\infty} u(y, x)f(y)\mu(dy), \quad f \in L^2, \tag{9}$$

which has eigenvalues $\{\lambda_i\}$ and eigenfunctions $\{g_i\}$ satisfying for each $i \geq 1$

$$\begin{cases} E[g_i(\xi_1)] = 0, \quad E\left[g_i^2(\xi_1)\right] = 1 \\ E[g_i(\xi_1)g_j(\xi_1)] = 0 \ (i \neq j), \quad E[u(\xi_1, x)g_i(\xi_1)] = \lambda_i g_i(x) \end{cases}. \tag{10}$$

With respect to (10), see Serfling (1980) [7], pp. 196 and Dunford and Schwartz (1963), pp. 905, 1009, 1083, 1087 for more details. Then we have

$$\lim_{n\to\infty} E\left[\left(u(\xi_i,\xi_j) - \sum_{k=1}^{n}\lambda_k g_k(\xi_i)g_k(\xi_j)\right)^2\right] = 0, \tag{11}$$

for each $i,j \geq 1$. Serfling (1980) [7] showed the non-central limit theorem for U-statistics with degree 2.

**Theorem 1.** *(Serfling (1980) [7])*
*Put $\theta = E[u(\xi_1,\xi_2)]$. Let $U_n$ be a U-statistics with the degenerate kernel $u(x,y)$ defined by*

$$U_n = \frac{2}{n^2} \sum_{1\leq i<j\leq n} u(\xi_i,\xi_j), \tag{12}$$

*Let $\{Z_i\}$ be i.i.d. random variables with the standard Normal distribution $N(0,1)$. Then, as $n \to \infty$*

$$nU_n \quad \Rightarrow \quad \sum_{j=1}^{\infty}\lambda_j\left(Z_j^2 - 1\right), \tag{13}$$

*where "$\Rightarrow$" means the weak convergence in* **R**.

It is well known that the rate of convergence in (13) is $O(n^{-1/2})$ (See, e.g., Serfling (1980) [7] for more details). We obtain asymptotic expansions for $U_n$ and $V_n$ using asymptotic expansions due to Sazonov—Uyanov (1995) [6] for sums of Hilbert space valued i.i.d. random variables in the next section.

### 4. Asymptotic Expansions for Single Sums which Hit a Ball in a Hilbert Space

In this section we consider an asymptotic expansions for sums of Hilbert space valued random vectors $\{X_i\}$ according to Sazonov—Uyanov (1995) [6]. Let $\{X_i\}$ be a sequence of i.i.d. random vectors in a separable Hilbert space $H$ with $E[X_1] = 0$ and $E\left[\|X_1\|^2\right] = 1$, where $\|x\|^2 = \langle x,x\rangle$ for $x \in H$ and $\langle\cdot,\cdot\rangle$ is the inner product in $H$. Define the covariance operator $V$ of $X_1$ by

$$\langle Vx,y\rangle = E[\langle X_1 - E[X_1],x\rangle\langle X_1 - E[X_1],y\rangle], \tag{14}$$

for $x,y \in H$. Denote by $\sigma_1^2 \geq \sigma_2^2 \geq \cdots$ the eigenvalues of $V$ and by $e_1,e_2,\cdots$ be the orthonormal eigenvectors corresponding to the eigenvalues. Put

$$S_n = \frac{1}{\sigma\sqrt{n}}\sum_{i=1}^{n}(X_i - E[X_i]), \quad v_k = \left(\prod_{i=1}^{k}\sigma_i\right)^{-1/k}, \quad c_k(V) = v_k^{k-1}, \tag{15}$$

where $\sigma^2 = E\left[\|X_1 - E(X_1)\|^2\right]$. Define the projection $K : H \to H$ by

$$Ky = \sum_{i=1}^{6k-5}\langle y,e_i\rangle e_i, \quad y \in H. \tag{16}$$

Put

$$\theta_k(L) = \sup\left\{\left|E\left[\exp\left(\sqrt{-1}\langle y,X_1\rangle\right)\right]\right| \,\bigg|\, \|Ky\| \geq \frac{1}{L}\right\}. \tag{17}$$

for any $L > 0$. Let $Y$ be the $H$-valued Gaussian random variables with mean 0 and the covariance operator $V$. For $a, h \in H$, $r > 0$, $i = 0, 1, \cdots$ we put

$$\Phi_i(a,r) = P\left\{\left\|\left(1 - \frac{i}{n}\right)^{1/2} Y - a\right\| < r\right\}, \tag{18}$$

$$d_h \Phi_i(a,r) = \lim_{t \to \infty} \frac{\Phi_i(a - th, r) - \Phi_i(a, r)}{t}. \tag{19}$$

Define the differential operators $d_h^k$ by

$$d_h^1 \Phi_i(a,r) = d_h \Phi_i(a,r), \quad d_h^k \Phi_i(a,r) = d_h\left(d_h^{k-1}\Phi_i(a,r)\right), \quad k \geq 2. \tag{20}$$

Put

$$\chi_{j,L}' = I\{\|X_j\| < L\} \tag{21}$$

for the indicator $I\{\cdot\}$

$$\chi_{j,t} = \chi_{j,\sqrt{n}(1+t)}{}' \tag{22}$$

and $\chi_j = \chi_{j,0}$. For positive integers $l_1, l_2, \cdots, l_s$ we put

$$Q_s = \left(d_{X_1\chi_1}^{l_1} - d_{Y_1}^{l_1}\right) \cdots \left(d_{X_1\chi_s}^{l_s} - d_{Y_s}^{l_s}\right) \tag{23}$$

and for integers $k \geq 2$, $1 \leq i \leq k - 2$, we put

$$A_i(a,r) = n^{-i/2} \sum_{j=1}^{n} {\sum}' n^{-j} \binom{n}{j} \left(l^{(j)}!\right)^{-1} E(Q_j)\Phi_j(a,r), \tag{24}$$

where $l^{(j)} = l_1! \cdots l_j!$ and $\sum'$ denotes the summation over all, such that

$$l_1 \geq 3, \ l_2 \geq 3, \ \cdots, \ l_j \geq 3, \ l_1 + \ l_2 + \cdots, +l_j = 2j + i. \tag{25}$$

The following theorem is the key result for the proofs of our theorems.

**Theorem 2.** *Sazonov—Uyanov (1995) [6]*
*Suppose that $E\left[\|X_1\|^p\right] < \infty$ for some $p \geq 4$. For any $t \geq 0$ and integer $k \geq 2$, let $L$ be a positive number, such that*

$$E\left[\|X_1\|^2\left(1 - \chi_{j,L}'\right)\right] \leq \frac{\sigma_{6k-5}^2}{3}. \tag{26}$$

*Then, for $L \leq n^{1/2}$*

$$\Delta_n(a,r) := \left|P\{\|S_n - a\| < r\} - P\{\|Y - a\| < r\} - \sum_{i=1}^{k-2} A_i(a,r)\right| \tag{27}$$

$$\leq A(p,s,t)$$

$$+ c(k)\exp\{-s^\alpha\}\left\{c_{6k-5}(V)E[B_2(a,r)(1 - \chi_1)] + \left(1 + M(a,r)^{k-2}\right)E[B_{k+1}(a,r)(1 - \chi_1)]\right.$$

$$\left. + c_{6k-5}(V)\left(1 + m^3(a,r)|a|\langle Va, a\rangle\right)^{k-2}\left(\frac{L^2}{n}\right)^{(k-1)/2} + \theta_k^{n/(k\log(n/L^2))}(L)\log\left(n/L^2\right)\right\},$$

*where for $s = |\|a\| - r|$ and $\alpha \geq \frac{1}{5}$,*

$$A(p,s,t) := nE[(1 - \chi_{1,t})] + c_p(1 - s)^p n^{1-p/2} E\left[\|X_1\|^p(\chi_{1,t} - \chi_1)\right], \tag{28}$$

$$B(j,r) = n^{-(j-2)}\left(\|X_1\|^j + m^j(a,r)|\langle X_1,a\rangle|^j\right),\tag{29}$$

$$M(a,r) = m^2(a,r)\langle Va,a\rangle\tag{30}$$

*and*

$$m(a,r) := \begin{cases} \min\left\{1,\frac{r}{\|a\|}\right\}, & \|a\| > 0 \\ 0, & a = 0 \end{cases}.\tag{31}$$

*In addition, the terms in the asymptotic expansion for $\varepsilon > 0$ satisfies the estimates*

$$|A_i(a,r)| \le c(\varepsilon,i)\exp\left\{-\frac{s^2}{2+\varepsilon}\right\}n^{-i/s}c_{6i+3}(V)\tag{32}$$

$$\times E\left[\chi_i\|X_1\|^{i+2} + |\langle X_1,a\rangle|^{i+2}\chi_i m^{i+2}(a,r)\times\right.$$

$$\left.\left\{1 + m^{2(i+2)}(a,r)\left(1 + m^{2(i+2)}(a,r)\langle Va,a\rangle^{i-1}\right)\right\} + M(a,r)^{3i+2}\right]$$

*for even i, and if i is odd, then we have*

$$|A_i(a,r)| \le c(\varepsilon,i)\exp\left\{-\frac{s^2}{2+\varepsilon}\right\}n^{-i/s}\left\{c_{6i+3}(V)\left(1 + \left(m^2(a,r)\langle Va,a\rangle^{i-1}\right)\right)\right.\tag{33}$$

$$\times E\left[|\langle X_1,a\rangle|\chi_i m(a,r)\left\{\|X_1\|^2 + \|X_1\|^{i+1} + \langle X_1,a\rangle^{i+1}(a,r)m^{i+1}(a,r)\right\}\right]$$

$$+ c_{6i+3}(V)m(a,r)\langle Va,a\rangle^{1/2}E\left[\chi_i\|X_1\|^{i+1}\right]$$

## 5. The Sato–Mercer Theorem

In the proofs of our theorems we use the Fourier series expansion for the kernel function $u(\xi_i,\xi_j)$ by eigenvalues and eigenfunction of the linear operator $T_u$ defined by (9). Since (11) holds in the sense of the $L^2$-convergence, (11) can not be applied to show the asymptotic expansions for U-statistics or V-statistics as it is. We show that $u(\xi_i,\xi_j)$ can be represented by the Fourier series expansion in (11) almost surely using the following Sato–Mercer theorem. (See Sato (1992) [8] for more details.)

**Theorem 3.** *(The Sato–Mercer theorem)*
    *Let X be a separable metric space with a Borel measure $\nu$ on X, and $K(x,y)$ be a function on $X \times X$ such that there exists a Borel-measurable subset $X_0$, such that*

$$\nu(X\backslash X_0) = 0.\tag{34}$$

*Suppose that $K(x,y)$ is continuous on $X_0$ and satisfies*

$$\int_X\int_X |K(x,y)|^2\nu(dx)\nu(dy) < \infty\tag{35}$$

*and*

$$\int_X\int_X K(x,y)f(x)\overline{f(y)}\nu(dx)\nu(dy) \ge 0,\tag{36}$$

*for any $f \in L^2(X,\nu)$. Then, the linear operator $T_K$ on $L^2(X,\nu)$ defined by*

$$T_K f(x) = \int_X K(x,y)f(y)\nu(dy), \quad f \in L^2(X,\nu)\tag{37}$$

*is nuclear if, and only if,*

$$\int_X K(x,x)\nu(dx) < \infty\tag{38}$$

*holds.*

From Theorem 3, we have the next result.

**Theorem 4.** *Let $\{\xi_j, \ j \geq 1\}$ be i.i.d. random variables with the distribution $\mu$. Let $u(x, y)$ be a real valued symmetric function on $\mathbf{R} \times \mathbf{R}$ and $T_u$ be a linear operator defined by*

$$T_u f(x) = E[u(\xi_1, x) f(\xi_1)] = \int_{-\infty}^{\infty} u(y, x) f(y) \mu(dy), \quad f \in L^2(\mathbf{R}, \mu). \tag{39}$$

*Suppose that $u(x, y)$ is the square integrable degenerate kernel of the linear operator $T_u$, such that*

$$\int_{-\infty}^{\infty} \int_{-\infty}^{\infty} u^2(x, y) \mu(dx) \mu(dy) < \infty, \tag{40}$$

$$\int_{-\infty}^{\infty} \int_{-\infty}^{\infty} u(x, y) f(x) \overline{f(y)} \mu(dx) \mu(dy) \geq 0, \tag{41}$$

*for any $f \in L^2(\mathbf{R}, \mu)$ and*

$$E[u(\xi_1, x)] = \int_{-\infty}^{\infty} u(y, x) \mu(dy) = 0 \tag{42}$$

*for any $x \in \mathbf{R}$. Let $\{\lambda_k\}$ and $\{g_k\}$ be eigenvalues and eingenfunctions of the linear operator $T_u$, respectively. Suppose*

$$\lambda_k \geq 0, \quad k \geq 1. \tag{43}$$

*Furthermore assume that there exists a Lebesgue measurable subset $X_0 \subset \mathbf{R}$, such that*

$$\mu(X_0) = 1 \tag{44}$$

*and $u(x, y)$ is continuous on $X_0$. Then, we have*

$$u(\xi_i, \xi_j) = \sum_{k=1}^{\infty} \lambda_k g_k(\xi_i) g_k(\xi_j) \quad a.s., \tag{45}$$

*for each $i, j \geq 1$.*

**Proof.** It is easy to see that from (10)

$$E\left[\sum_{k=1}^{n} |\lambda_k g_k(\xi_i) g_k(\xi_j)|\right] = \sum_{k=1}^{n} E\left[|\lambda_k g_k(\xi_i) g_k(\xi_j)|\right] \tag{46}$$

$$= \sum_{k=1}^{n} |\lambda_k| E\left[|g_k(\xi_i) g_k(\xi_j)|\right]$$

$$\leq \sum_{k=1}^{n} |\lambda_k| \left\{E\left[g_k(\xi_i)^2\right]\right\}^{1/2} \left\{E\left[g_k(\xi_j)^2\right]\right\}^{1/2}$$

$$= \sum_{k=1}^{n} |\lambda_k|,$$

for each $n \geq 1$. Tending $n \to \infty$, (46) implies that

$$E\left[\sum_{k=1}^{\infty} |\lambda_k g_k(\xi_i) g_k(\xi_j)|\right] \leq \sum_{k=1}^{\infty} |\lambda_k|. \tag{47}$$

On the other hand, from (40) and (41), $u(x, y)$ satisfies (35) and (36). Therefore, $T_u$ is nuclear by Theorem 3. Hence, from (43) and the nuclearity of $T_u$, we have

$$\sum_{k=1}^{\infty} |\lambda_k| = \sum_{k=1}^{\infty} \lambda_k < \infty. \tag{48}$$

From (47) and (48), we have

$$E\left[\sum_{k=1}^{\infty} |\lambda_k g_k(\xi_i) g_k(\xi_j)|\right] < \infty, \tag{49}$$

which implies

$$\sum_{k=1}^{\infty} |\lambda_k g_k(\xi_i) g_k(\xi_j)| < \infty, \quad a.s. \tag{50}$$

Therefore, (45) is proved from (11) and (50).   □

**Remark 1.** *If the symmetric function $u(x, y)$ is piecewise continuous on $\mathbf{R}$, then there exists $X_0 \subset \mathbf{R}$ satisfying (44) such that $u(x, y)$ is continuous on $X_0$. In the next section, we show a typical example of U- or V-statistics defined by such piecewise continuous function $u(x, y)$ as its kernel function.*

## 6. Asymptotic Expansions for Degenerate V-Statistics and U-Statistics with Degree 2

For applying Theorem 2 for Hilbert space valued random variables to the proof of asymptotic expansions for $V_n$, we represent $V_n$ by sums of Hilbert space valued random variables $\{G_i\}$ by the following method.

According to K.—Yoshihara (1994) [9], we introduce a separable Hilbert space $H$-equipped with the inner product $\langle \cdot, \cdot \rangle$ and the norm $\| \cdot \|$ as follows,

$$H = \left\{\mathbf{x} = (x_1, x_2, \cdots) \in \mathbf{R}^{\infty} \,\middle|\, \sum_{k=1}^{\infty} |\lambda_k| x_k^2 < \infty\right\}, \tag{51}$$

$$\langle \mathbf{x}, \mathbf{y} \rangle = \sum_{k=1}^{\infty} |\lambda_k| x_k y_k \tag{52}$$

and

$$\|\mathbf{x}\| = \left(\sum_{k=1}^{\infty} |\lambda_k| x_k^2\right)^{1/2}. \tag{53}$$

Using the assumptions of Theorem 4, we have from (10) and (48) that

$$E\left[\sum_{k=1}^{\infty} |\lambda_k| g_k^2(\xi_i)\right] = \sum_{k=1}^{\infty} |\lambda_k| E\left[g_k^2(\xi_i)\right] = \sum_{k=1}^{\infty} |\lambda_k| < \infty, \tag{54}$$

which implies that we can define $H$-valued random variables by

$$G_i = (g_1(\xi_i), g_2(\xi_i), g_3(\xi_i), \cdots) \tag{55}$$

for each $i \geq 1$. Let $\{U_n, \ n \geq 1\}$ and $\{V_n, \ n \geq 1\}$ be U-statistics and V-statistics with degree 2 defined by (3), respectively.

**Theorem 5.** *Without loss of generality we assume that $\theta = 0$. Suppose that $\{\xi_j, \ j \geq 1\}$ is a sequence of i.i.d. random variables with the distribution $\mu$. Assume that $u(x, y)$ is a square integrable symmetric function with respect to $\mu \times \mu$ satisfying (40) $\sim$ (42). Suppose that for some $p \geq 4$*

$$E\left[\|G_1\|^p\right] < \infty. \tag{56}$$

*Furthermore, without loss of generality, assume that*

$$\sum_{k=1}^{\infty} \lambda_k = 1. \tag{57}$$

*Let $Y$ be the H-valued Gaussian random variables with mean 0 and the covariance operator $V$ satisfying (14) with the eigenvalues $\sigma_1^2 \geq \sigma_2^2 \geq \cdots$ and the orthogonal eigenvectors $e_1, e_2, \cdots$. For any $t \geq 0$, integer $k \geq 2$, let $L$ be a positive number, such that*

$$E\left[\|G_1\|^2 \left(1 - \chi_{j,L'}\right)\right] \leq \frac{\sigma_{6k-5}^2}{3}. \tag{58}$$

*Then, for $L \leq n^{1/2}$ and $\alpha \geq \frac{1}{5}$,*

$$\left| P\{|nV_n| \leq r\} - P\{\|Y\| \leq r\} - \sum_{i=1}^{k-2} A_i(0, r) \right| \tag{59}$$

$$\leq A(p, s, t) + c(k) \exp\{-r^\alpha\} [c_{6k-5}(V) E[B_2(0, r)(1 - \chi_1)]$$

$$+ E[B_{k+1}(0, r)\chi_1] + c_{6k-5}(V) \left(\frac{L^2}{n}\right)^{(k-1)/2} + \theta_k^{n/(k \log(n/L^2))}(L) \log\left(n/L^2\right),$$

*where*

$$\|Y\| = \left| \sum_{j=1}^{\infty} \lambda_j \left(Z_j^2 - 1\right) \right|, \tag{60}$$

$$A(p, s, t) = nE[(1 - \chi_{1,t})] + c(p)(1 + r)^{-p} n^{1-p/2} E\left[\|G_1\|^p (\chi_{1,t} - \chi_1)\right] \tag{61}$$

*and*

$$B_j(0, r) = n^{-(j-2)/2} \|G_1\|^j. \tag{62}$$

**Proof.** Put

$$h(\mathbf{x}) = \sum_{k=1}^{\infty} \lambda_k x_k \tag{63}$$

for

$$\mathbf{x} \in H = \left\{ \mathbf{x} = (x_1, x_2, \cdots) \,\middle|\, \sum_{k=1}^{\infty} |\lambda_k| x_k^2 < \infty \right\} \tag{64}$$

Recall that for each $i$,

$$\frac{1}{\sqrt{n}} \sum_{i=1}^{n} G_i = \left( \frac{1}{\sqrt{n}} \sum_{i=1}^{n} g_1(\xi_i), \frac{1}{\sqrt{n}} \sum_{i=1}^{n} g_2(\xi_i), \cdots \right) \in H. \tag{65}$$

Then we have

$$nV_n = \frac{1}{n} \sum_{1 \leq i,j \leq n} u\left(\xi_i, \xi_j\right) = \frac{1}{n} \sum_{1 \leq i,j \leq n} \sum_{k=1}^{\infty} \lambda_k g_k(\xi_i) g_k(\xi_j) \tag{66}$$

$$= \frac{1}{n} \sum_{k=1}^{\infty} \lambda_k \left\{ \sum_{1 \leq i,j \leq n} g_k(\xi_i) g_k(\xi_j) \right\} = \frac{1}{n} \sum_{k=1}^{\infty} \lambda_k \left\{ \sum_{i=1}^{n} g_k(\xi_i) \right\}^2$$

$$= \left\| \frac{1}{\sqrt{n}} \sum_{i=1}^{n} G_i \right\|.$$

Thus, we can apply Theorem 2 to show Theorem 5. $\square$

**Theorem 6.** *Suppose that the i.i.d. random variables $\{\xi_i,\ i \geq 1\}$ obey a continuous distribution. Let $v(x,y)$ be a symmetric function defined by*

$$v(x,y) = \begin{cases} u(x,y), & x \neq y \\ 0, & x = y \end{cases}. \tag{67}$$

*Under the same assumptions in Theorem 5, the equation (59) holds for $U_n$ with the degenerate kernel $v(x,y)$.*

**Proof.** Since the i.i.d. random variables $\{\xi_i,\ i \geq 1\}$ obey a continuous distribution, we have

$$P\{\xi_i \neq \xi_j\} = 1 \quad (i \neq j). \tag{68}$$

Therefore, from (67) and (68)

$$nU_n = \frac{2}{n} \sum_{1 \leq i < j \leq n} u(\xi_i, \xi_j) = \frac{1}{n} \sum_{1 \leq i,j \leq n} v(\xi_i, \xi_j) \quad a.s. \tag{69}$$

Since the right hand side of (69) is the V-statistics with the degenerate kernel $v(x,y)$ satisfying all assumptions of Theorem 5, Theorem 6 holds from Theorem 5.　□

**Remark 2.** *From (10), $E[G_1] = 0$ and $\sigma^2 = E\left[\|G_1\|^2\right] = 1$ in Theorem 5.*

## 7. Cramer–Von Mises Statistics

There are some examples of U-statistics or V-statistics for which the above theorems are applicable under the assumption of nuclearity of the kernel functions where the above theorems are applicable.

**Example 3.** *(Cramér-von Mises Statistics, Sato (1992) [8])*
*Assume that i.i.d. random variables $\{\xi_j,\ j \geq 1\}$ obey the uniform distribution $U(0,1)$, i.e., $\mu$ is the Lebesgue on $[0,1]$. Define a kernel function $u(x,y)$ by*

$$u(x,y) = \int_0^1 \frac{\left(I_{[x,1]}(t) - t\right)\left(I_{[y,1]}(t) - t\right)}{t(1-t)} dt, \quad x,y \in [0,1] \tag{70}$$

*satisfies the hypothesis of Theorem 5 or Theorem 6. On the other hand, we have*

$$\int_0^1 u(x,x)dx = \int_0^1 dx \int_0^1 \frac{\left(I_{[x,1]}(t) - t\right)^2}{t(1-t)} dt \tag{71}$$

$$= \int_0^1 \frac{dt}{t(1-t)} \int_0^1 \left(I_{[x,1]}(t) - t\right)^2 dx = 1 < \infty.$$

*Therefore, the integral operator $T_u$ defined by*

$$T_u f(y) = \int_0^1 u(x,y)f(y)dx \tag{72}$$

*is nuclear from Theorem 3. Therefore, since the degenerate kernel $u(x,y)$ defined by (70) satisfies all assumptions of Theorem 5, Theorem 5 holds for the Cramér-von Mises Statistics. Furthermore, Theorem 6 also holds for U-statistics with the degenerate kernel $v(x,y)$ defined by (67) and (70).*

## 8. Conclusions

Bentkus—Götze (1999) [4] and Zubayraev (2011) [5] obtained the remainder $O(n^{-1})$ in asymptotic expansions for U-statistics or V-statistics with degenerate kernels. From Theorems 5 and 6, if we assume $E\left[\|G_1\|^p\right] \leq \infty$, $p \geq 4$ and some conditions, then we

obtain the remainder $O\left(n^{1-p/2}\right)$. Applying Theorem 5, we obtain asymptotic expansions for the Cramér–von Mises statistics of the uniform distribution $U(0,1)$ with the remainder $O\left(n^{1-p/2}\right)$ for any $p \geq 4$.

**Funding:** Grant-in-Aid Scientific Research (C), No.18K03431, Ministry of Education, Science and Culture, Japan.

**Institutional Review Board Statement:** Not applicable.

**Informed Consent Statement:** Not applicable.

**Data Availability Statement:** The author uses no data.

**Acknowledgments:** The author would like to express his gratitude to the anonymous referees for their useful comments. He also would like to express his gratitude to V.V. Ulyanov for giving the opportunity to present this work.

**Conflicts of Interest:** The author declares no conflict of interest.

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
