# Peer review of "Asymptotic Expansions for Symmetric Statistics with Degenerate Kernels"

_mathematics, doi:10.3390/math10214158_

Round 1
Reviewer 1 Report
Comments and suggestions for the author are in the file

Author Response
Thank you very much for your message.
Please see red parts.

Reviewer 2 Report
1) Revise the paper according to Paper Submission Guide;
2) We need some paragraphs to make a clear introduction of your case.
3) Connected between introduction and the next section must be clear. Relation between equation 1-2 and 3
4) Section 2 about SYMMETRIC STATISITCS, we need more clear regarding symetric statistics related to equation (4).
5) in Equation (10), is that true?
6) Please check again for equation (27)
7) Since the right hand side of (69) is the V-statistics with the degenerate kernel v(x; y) satisfying all assumptions of Theorem 5, Theorem 6 holds from Theorem 5.
(69)??
8) Complete the Response to the Comments, and send to us, along with the revised manuscript.
Author Response
Thank you for your message.
Please the red parts.

Round 2
Reviewer 2 Report
(1) Could you please add some references especially come from this journal (mathematics-mdpi)!
(2) Equation (34) from Equation (11), can you add one sentence, why?
(3) The conclusions can be improved related to the results, I can not see the comprehensive conclusion?
Author Response
- Could you please ….
Please see “References” in pp.10. I added “[https://doi.org/10. …]
- Equation (34) from Equation (11), can you add one sentence, why?
I erased the equation (34). Please see the blue sentences in pp. 5.
- The conclusion can be improved related to the results, I can not see the comprehensive conclusion?
I added the section 8 “Conclusion”. Please see blue sentences in pp. 10.
